# CONFORMAL PREDICTION MASKS: VISUALIZING UNCERTAINTY IN MEDICAL IMAGING

**Gilad Kutiel, Regev Cohen, Michael Elad, Daniel Freedman & Ehud Rivlin**
Verily Research, Israel.
`regevcohen@google.com`

## ABSTRACT

Estimating uncertainty in image-to-image recovery networks is an important task, particularly as such networks are being increasingly deployed in the biological and medical imaging realms. A recent conformal prediction technique derives per-pixel uncertainty intervals, guaranteed to contain the true value with a user-specified probability. Yet, these intervals are hard to comprehend and fail to express uncertainty at a conceptual level. In this paper, we introduce a new approach for uncertainty quantification and visualization, based on masking. The proposed technique produces interpretable image masks with rigorous statistical guarantees for image regression problems. Given an image recovery model, our approach computes a mask such that a desired divergence between the masked reconstructed image and the masked true image is guaranteed to be less than a specified risk level, with high probability. The mask thus identifies reliable regions of the predicted image while highlighting areas of high uncertainty. Our approach is agnostic to the underlying recovery model and the true unknown data distribution. We evaluate the proposed approach on image colorization, image completion, and super-resolution tasks, attaining high quality performance on each.

## 1 INTRODUCTION

Deep Learning has been successful in many applications, spanning computer vision, speech recognition, natural language processing, and beyond (Cohen et al., 2021a;b). For many years, researchers were mainly content in developing new techniques that achieve unprecedented accuracy, without concerns for understanding the uncertainty implicit in such models. Recently, however, there has been a concerted effort within the research community to quantify the uncertainty of deep models.

This paper addresses the problem of quantifying and visualizing uncertainty in the realm of image-to-image tasks. Such problems include super-resolution, deblurring, colorization, and image completion, amongst others. Assessing uncertainty is important generally, but is particularly so in application domains such as biological and medical imaging, in which fidelity to the ground truth is paramount. If there is an area of the reconstructed image where such fidelity is unlikely or unreliable due to high uncertainty, this is crucial to convey.

Our approach to uncertainty estimation is based on masking. Specifically, we are interested in computing a mask such that the uncertain regions in the image are masked out. Based on conformal prediction (Angelopoulos & Bates, 2021a), we derive an algorithm that can apply to any existing image-recovery model and produce uncertainty a mask satisfying the following criterion: the divergence between the masked reconstructed image and the masked true image is guaranteed to be less than a specified level, with high probability. The resultant mask highlights areas in the recovered image of high uncertainty while trustworthy regions remain intact. Our distribution-free method, illustrated in Figure 1, is agnostic to the prediction model and to the choice of divergence function, which should be dictated by the application. Our contributions are as follows:

1. We introduce the notion of conformal prediction masks: a distribution-free approach to uncertainty quantification in image-to-image regression. We derive masks which visually convey regions of uncertainty while rigorously providing strong statistical guarantees for any regression model, image dataset and desired divergence measure.

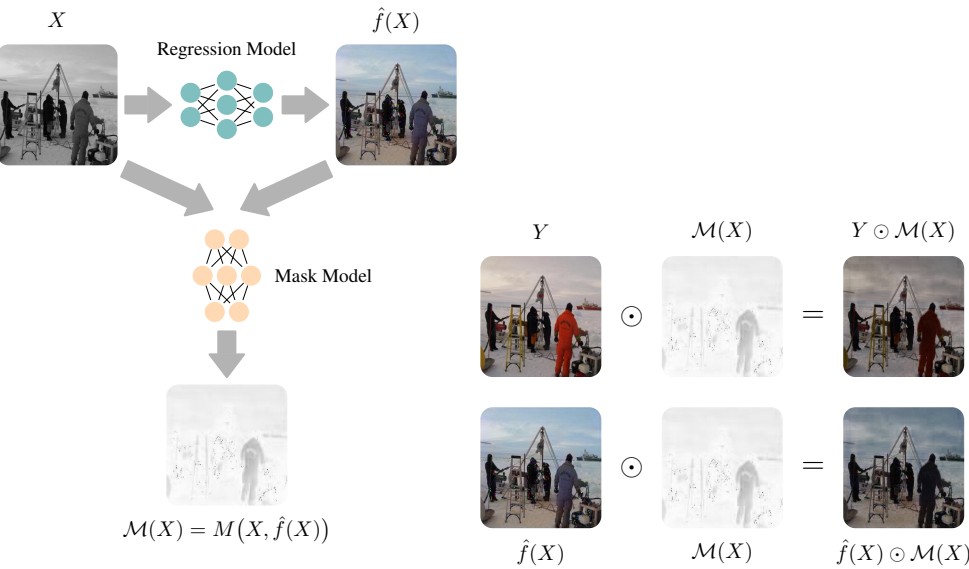

Figure 1: **High-level overview**. Given image measurements $X$ (e.g gray-scale image) of a ground-truth image $Y$, and a predicted image $\hat{f}(X)$ (e.g. colorized image), the mask model outputs an uncertainty mask $\mathcal{M}(X)$ such that the divergence between the masked ground-truth and the masked prediction is below a chosen risk level with high probability.

2. We develop a practical training algorithm for computing these masks which only requires triplets of input (degraded), reconstructed and true images. The resultant mask model is trained once for all possible risk levels and is calibrated via a simple process to meet the required guarantees given a user-specified risk level and confidence probability.

3. We demonstrate the power of the method on image colorization, image completion and super-resolution tasks. By assessing our performance both visually and quantitatively, we show the resultant masks attain the probabilistic guarantee and provide interpretable uncertainty visualization without over-masking the recovered images, in contrast to competing techniques.

## 2 RELATED WORK

**Bayesian Uncertainty Quantification** The Bayesian paradigm defines uncertainty by assuming a distribution over the model parameters and/or activation functions. The most prevalent approach is Bayesian neural networks (MacKay, 1992; Valentin Jospin et al., 2020; Izmailov et al., 2020), which are stochastic models trained using Bayesian inference. Yet, as the number of model parameters has grown rapidly, computing the exact posteriors has became computationally intractable. This shortcoming has led to the development of approximation methods such as Monte Carlo dropout (Gal & Ghahramani, 2016; Gal et al., 2017a), stochastic gradient Markov chain Monte Carlo (Salimans et al., 2015; Chen et al., 2014), Laplacian approximations (Ritter et al., 2018) and variational inference (Blundell et al., 2015; Louizos & Welling, 2017; Posch et al., 2019). Alternative Bayesian techniques include deep Gaussian processes (Damianou & Lawrence, 2013), deep ensembles (Ashukha et al., 2020; Hu et al., 2019), and deep Bayesian active learning (Gal et al., 2017b), to name just a few. A comprehensive review on Bayesian uncertainty quantification is given in Abdar et al. (2021).

**Distribution-Free Methods and Conformal Prediction** Unlike Bayesian methods, the frequentist approach assumes the true model parameters are fixed with no underlying distribution. Examples of such distribution-free techniques are model ensembles (Lakshminarayanan et al., 2017; Pearce et al., 2018), bootstrap (Kim et al., 2020; Alaa & Van Der Schaar, 2020), interval regression (Pearce et al., 2018; Kivaranovic et al., 2020; Wu et al., 2021) and quantile regression (Gasthaus et al., 2019; Romano et al., 2019). An important distribution-free technique which is most relevant to our work is conformal prediction (Angelopoulos & Bates, 2021b; Shafer & Vovk, 2008). This approach relies on a labeled calibration dataset to convert point estimations into prediction regions. Conformal meth-

ods can be used with any estimator, require no retraining, are computationally efficient and provide coverage guarantees in finite samples (Lei et al., 2018). Recent development includes conformalized quantile regression (Romano et al., 2019; Sesia & Candès, 2020; Angelopoulos et al., 2022b), conformal risk control (Angelopoulos et al., 2022a; Bates et al., 2021; Angelopoulos et al., 2021) and semantic uncertainty intervals for generative adversarial networks (Sankaranarayanan et al., 2022). Sun (2022) provides an extensive survey on distribution-free conformal prediction methods.

## 3 BACKGROUND: CONFORMAL PREDICTION IN IMAGE REGRESSION

We present a brief overview of the work in (Angelopoulos et al., 2022b), which stands out in the realm of conformal prediction for image-to-image problems, and serves as the basis of our work. Let $Y \in \mathcal{Y} = \mathbb{R}^N$ be a ground-truth image in vector form, an image $X \in \mathcal{X} = \mathbb{R}^M$ be its measurements, and $\hat{f}(X) \in \mathcal{Y}$ an estimator of $Y$. Conformal prediction constructs uncertainty intervals

$$\mathcal{T}(X)_{[i]} = \left[ \hat{f}(X)_{[i]} - \hat{l}(X)_{[i]}, \hat{f}(X)_{[i]} + \hat{u}(X)_{[i]} \right], \quad i = 0, ..., N-1, \tag{1}$$

where $\hat{l}(X)_{[i]} \geq 0$ and $\hat{u}(X)_{[i]} \geq 0$ represent the uncertainty in lower and upper directions respectively. Given heuristic uncertainty values $\tilde{l}$ and $\tilde{u}$, the uncertainty intervals are calibrated using a calibration dataset $\mathcal{C} \triangleq \{X_k, Y_k\}_{k=1}^K$ to guarantee they contain at least a fraction $\alpha$ of the ground-truth pixel values with probability $1 - \delta$. Here $\alpha \in (0, 1)$ and $\delta \in (0, 1)$ are user-specified risk and error levels respectively. Formally, the per-pixel uncertainty intervals are defined as follows.

**Definition 1.** *Risk-Controlling Prediction Set (RCPS). A random set-valued function $\mathcal{T} : \mathcal{X} \to \mathcal{Y}' = 2^{\mathcal{Y}}$ is an $(\alpha, \delta)$-Risk-Controlling Prediction Set if*

$$\mathbb{P}(\mathcal{R}(\mathcal{T}) \leq 1 - \alpha) \geq 1 - \delta.$$

*Here the risk is $\mathcal{R}(\mathcal{T}) \triangleq 1 - \mathbb{E}\left[ \frac{1}{N} \left| \{i \,:\, Y_{[i]}^{test} \in \mathcal{T}(X^{test})_{[i]}\} \right| \right]$ where the expectation is over a new test point $(X^{test}, Y^{test})$, while the outer probability is over the calibration data.*

The procedure for constructing RCPS consists of two stages. First, a machine learning system (e.g. neural network) is trained to output a point prediction $\hat{f}$, and heuristic lower and upper interval widths $(\tilde{l}, \tilde{u})$. The second phase utilizes the calibration set to calibrate $(\tilde{l}, \tilde{u})$ so they contain the right fraction of ground truth pixels. The final intervals are those in (1) with the calibrated widths $(\hat{l}, \hat{u})$.

Conformal prediction provides per-pixel uncertainty intervals with statistical guarantees in image-to-image regression problems. Yet, the per-pixel prediction sets may be difficult to comprehend on their own. To remedy this, the uncertainty intervals are visualized by passing the pixel-wise interval lengths through a colormap, where small sets render a pixel blue and large sets render it red. Thus, the redder a region is, the greater the uncertainty, and the bluer it is, the greater the confidence. The resultant uncertainty map, however, is not directly endowed with rigorous guarantees. This raises the following question: *can we directly produce an uncertainty map with strong statistical guarantees?*

## 4 CONFORMAL PREDICTION MASKS

Inspired by the above, we construct uncertainty masks $\mathcal{M}(X) = M(X, \hat{f}(X)) \in [0, 1]^N$ such that

$$\mathbb{E}\left[ \mathcal{M}(X^{test})_{[i]} \cdot \left| \hat{f}(X^{test})_{[i]} - Y_{[i]}^{test} \right| \right] \leq \beta_{[i]}, \tag{2}$$

where the expectation is over a new test point, and $\beta_{[i]} \in \mathbb{R}^+$ is user-specified risk level. Define $\hat{f}_{\mathcal{M}}(X) \triangleq \mathcal{M}(X) \odot \hat{f}(X)$ and $Y_{\mathcal{M}} \triangleq \mathcal{M}(X) \odot Y$ where $\odot$ represents a point-wise (Hadamard) product. Then, note that building (2) is equivalent to create the following uncertainty intervals

$$\mathcal{T}_{\mathcal{M}}(X)_{[i]} = \left[ \hat{f}_{\mathcal{M}}(X)_{[i]} - \beta_{[i]}, \hat{f}_{\mathcal{M}}(X)_{[i]} + \beta_{[i]} \right], \tag{3}$$

which satisfies

$$Y_{\mathcal{M}[i]}^{test} \in \mathcal{T}_{\mathcal{M}}(X^{test})_{[i]}. \tag{4}$$

We remark a few difference between (3) and (1): In (1) the lower and upper per-pixel uncertainty widths $(\hat{l}, \hat{u})$ depend on $X$ and are calibrated, while in (3) $\hat{l} = \hat{u} \equiv \beta$ are user-specified and independent of $X$. Furthermore, the uncertainty parameters which undergo calibration are $\{\mathcal{M}(X)_{[i]}\}_{i=1}^{N}$.

One may notice that the above formulation exhibits a major limitation as each value of the prediction mask is defined independently from other values. Hence, it requires the user to specify a risk level for each pixel which is cumbersome, especially in high dimension. More importantly, setting each entry of the mask independently may fail in capturing the dependency between pixels, thus, fail to express uncertainty at a conceptual level. To overcome this, we redefine our uncertainty masks to ensure with probability at least $1 - \delta$ it holds that $\mathbb{E}\left[\left\|\hat{f}_{\mathcal{M}}(X^{\text{test}}) - Y_{\mathcal{M}}^{\text{test}}\right\|_1\right] \leq \alpha$, where $\alpha \in \mathbb{R}^+$ is a global risk level and $\|Z\|_1 \triangleq \sum_{i=1}^{N} Z[i]$ is the L1 norm of an arbitrary image $Z$. Furthermore, the latter formulation can be generalized to any divergence measure $d : \mathcal{Y} \times \mathcal{Y} \to \mathbb{R}^+$ such that

$$\mathbb{E}\left[d\left(\hat{f}_{\mathcal{M}}(X^{\text{test}}), Y_{\mathcal{M}}^{\text{test}}\right)\right] \leq \alpha. \tag{5}$$

Note we avoid trivial solutions, e.g. a zero-mask, which satisfy (5) yet provide no useful information. Thus, we seek solutions that employ the least masking required to meet (5), with high probability.

The above formulation enjoys several benefits. First, the current definition of the mask captures pixel-dependency. Thus, rather than focusing on individual pixels, the resultant map would mask out (or reduce) regions of high uncertainty within the predicated image to guarantee the divergence remains below the given risk level. Second, it accepts any divergence measure, each leading to a different mask. For example, selecting $d(\cdot, \cdot)$ to be a distortion measure may underline uncertainty regions of high-frequency objects (e.g. edges), while setting $d(\cdot, \cdot)$ to be a perceptual loss may highlight semantic factors within the image. Formally, we refers to these uncertainty masks as *Risk-Controlling Prediction Masks*, which are defined below.

**Definition 2.** *Risk-Controlling Prediction Mask (RCPM).* A random function $\mathcal{M} : \mathcal{X} \times \mathcal{Y} \to [0, 1]^{\mathcal{Y}}$ is an $(\alpha, \delta)$-Risk-Controlling Prediction Mask if

$$\mathbb{P}\left(\mathbb{E}\left[\mathcal{R}(\mathcal{M})\right] \leq \alpha\right) \geq 1 - \delta,$$

where the risk is defined as $\mathcal{R}(\mathcal{M}) \triangleq d\left(\hat{f}_{\mathcal{M}}(X^{test}), Y_{\mathcal{M}}^{test}\right)$ for given a divergence $d(\cdot, \cdot)$. The outer probability is over the calibration data, while the expectation taken over a test point $(X^{test}, Y^{test})$.

As for RCPS, the procedure for creating RCPM includes two main stages. First, given a predictor $\hat{f}$, we require a heuristic notion of a non-zero uncertainty mask $\widetilde{M}$. In particular, we train a neural network to output a mask given the measurements and the predicted image as inputs. Second, given a divergence measure, we use the calibration set to calibrate the heuristic mask until the divergence measure decreases below the desired risk level. The final outputs are the calibrated mask and the original prediction multiplied by the mask. The overall method is outlined in Algorithm 1. Following the latter, we now discuss notion of initial uncertainty masks and the subsequent calibration process.

---

**Algorithm 1** Generating RCPM

---

1. Given a regression model $\hat{f}$, train a model which outputs an initial mask $\widetilde{\mathcal{M}}$.

2. Calibrate $\widetilde{\mathcal{M}}$ using the calibration dataset to obtain $\mathcal{M}$ (e.g. using Algorithm 2).

3. Given $X$ at inference, output the risk-controlling masked prediction $\hat{f}_{\mathcal{M}}(X) = \mathcal{M}(X) \odot \hat{f}(X)$.

---

## 4.1 Initial Estimation of Uncertainty Masks

Here we present two notions of uncertainty masks. The first concept, based on (Angelopoulos et al., 2022b), translates given uncertainty intervals into a heuristic mask. In the second we develop a process for training a neural network which accepts the input and the predicted images and outputs an uncertainty mask based on a given divergence between the prediction and the ground-truth image.

### 4.1.1 Intervals to Masks

In (Angelopoulos et al., 2022b), the authors propose to build uncertainty intervals based on four heuristic notions of lower and upper interval widths $\tilde{l}$ and $\tilde{u}$: (1) Regression to the magnitude of the residual; (2) one Gaussian per pixel; (3) softmax outputs; and (4) pixel-wise quantile regression. Then, we build a mask by setting the pixel-values to be inversely proportional to the interval sizes:

$$\widetilde{\mathcal{M}}(X)_{[i]} \propto \left(\tilde{u}_{[i]} - \tilde{l}_{[i]}\right)^{-1}. \tag{6}$$

Thus, the resultant mask holds high values at pixels with small-size intervals (high confidence) and smaller values at pixels with larger intervals corresponding to high uncertainty regions. However this approach requires first creating uncertainty intervals , hence, we next introduce a technique which directly produces an uncertainty mask.

### 4.1.2 Mask Regression

Here, we introduce a notion of an uncertainty mask represented by a neural network $\widetilde{\mathcal{M}}(X;\theta) \in [0,1]^N$ with parameters $\theta$. The mask model is trained to output a mask which satisfies

$$\mathbb{E}\left[d\left(\hat{f}_{\widetilde{\mathcal{M}}}(X^{\text{train}}), Y^{\text{train}}_{\widetilde{\mathcal{M}}}\right)\right] \leq \alpha. \tag{7}$$

where here the expectation is over the training samples $\mathcal{D} \triangleq \{X_j, Y_j\}_{j=1}^{J}$ used to train $\hat{f}$. To derive our loss function, we start with formulating the following problem for a given a triplet $\left(X, Y, \hat{f}(X)\right)$

$$\min_{\theta} \; ||\widetilde{\mathcal{M}}\left(X, \hat{f}(X)\right) - \mathbb{1}||_2^2 \quad \text{subject to} \quad d\left(\hat{f}_{\widetilde{\mathcal{M}}}(X), Y_{\widetilde{\mathcal{M}}}\right) \leq \alpha, \tag{8}$$

where $\mathbb{1}$ is an image of all ones, representing no masking. The constraint in the above corresponds to (7), while the objective aims to find the minimal solution, i.e., the solution that masks the image the least (avoding trivial solutions). The Lagrangian of the problem is given by

$$\mathcal{L}(\theta, \mu) \triangleq ||\widetilde{\mathcal{M}}\left(X, \hat{f}(X)\right) - \mathbb{1}||_2^2 + \mu\left(d\left(\hat{f}_{\widetilde{\mathcal{M}}}(X), Y_{\widetilde{\mathcal{M}}}\right) - \alpha\right) \tag{9}$$

where $\mu > 0$ is the dual variable, considered as an hyperparameter. Given $\mu$, the optimal mask can be obtained by minimizing $\mathcal{L}(\theta, \mu)$ with respect to $\theta$, which is equivalent to minimizing

$$||\widetilde{\mathcal{M}}\left(X, \hat{f}(X)\right) - \mathbb{1}||_2^2 + \mu \cdot d\left(\hat{f}_{\widetilde{\mathcal{M}}}(X), Y_{\widetilde{\mathcal{M}}}\right) \tag{10}$$

since $\alpha$ does not depend on $\theta$. Thus, we train our mask model using the following loss function:

$$\mathcal{L}(\mathcal{D}, \theta) \triangleq \sum_{(X,Y) \in \mathcal{D}} ||\widetilde{\mathcal{M}}\left(X, \hat{f}(X)\right) - \mathbb{1}||_2^2 + \mu \cdot d\left(\hat{f}_{\widetilde{\mathcal{M}}}(X), Y_{\widetilde{\mathcal{M}}}\right). \tag{11}$$

The proposed approach facilitates the use of any differentiable distortion measure and is agnostic to the prediction model $\hat{f}$. Furthermore, notice that the loss function is independent of $\alpha$, hence, can be trained once for all values of $\alpha$. Thus, the output mask acts only as an initial uncertainty map which may not satisfy (5) and need to be calibrated. Following proper calibration, discussed next, our mask model attains (5) without requiring the ground-truth $Y$. Lastly, this approach directly outputs uncertainty masks and thus it is the focus of our work.

### 4.2 Mask Calibration

We consider the $\widetilde{M}(X)$ as an initial estimation of our uncertainty mask which needs to calibrated to provide the guarantee in Definition 2. As the calibration process is not the focus of our work, we perform a simple calibration outlined in Algorithm 2. The core of the calibration employs a parametric function $C(\cdot; \lambda)$ pixel-wise to obtain a mask $\mathcal{M}_{\lambda}(X)_{[i]} \triangleq C\left(\widetilde{\mathcal{M}}(X)_{[i]}; \lambda\right)$. In general, $C(\cdot; \lambda)$ can be any monotonic non-decreasing function. Here we consider the following form[1]

$$\mathcal{M}_{\lambda}(X)_{[i]} \triangleq \min\left(\frac{\lambda}{1 - \widetilde{\mathcal{M}}(X)_{[i]} + \epsilon}, 1\right) \quad \forall i = 1, ..., N, \tag{12}$$

---

[1]A small value $\epsilon$ is added to the denominator to ensure numerical stability.

which has been found empirically to perform well in our experiments. To set $\lambda > 0$, we use the calibration dataset $\mathcal{C} \triangleq \{X_k, Y_k\}_{k=1}^K$ such that for any pair $(X_k, Y_k) \in \mathcal{C}$ we compute

$$\lambda_k \triangleq \max \left\{ \hat{\lambda} : d\left(\hat{f}_{\mathcal{M}_{\hat{\lambda}}}(X_k), Y_{k\,\mathcal{M}_{\hat{\lambda}}}\right) \leq \alpha \right\}. \tag{13}$$

Finally, $\lambda$ is taken to be the $1 - \delta$ quantile of $\{\lambda_k\}_{k=1}^K$, i.e. the maximal value for which at least $\delta$ fraction of the calibration set satisfies condition (5). Thus, assuming the calibration and test sets are i.i.d samples from the same distribution, the calibrated mask is guaranteed to satisfy Definition 2.

---

**Algorithm 2** Calibration Process

---

**Input:** Calibration data $\mathcal{C} \triangleq \{X_k, Y_k\}_{k=1}^K$; risk level $\alpha$; error rate $\delta$; underlying predictor $\hat{f}$; heuristic mask $\widetilde{\mathcal{M}}$; a monotonic non-decreasing function $C(\cdot; \lambda) : [0, 1] \to [0, 1]$ parameterized by $\lambda > 0$.

1. For a given $\tilde{\lambda} > 0$, define $\mathcal{M}_{\tilde{\lambda}}(X)_{[i]} \triangleq C\left(\widetilde{\mathcal{M}}(X)_{[i]}; \tilde{\lambda}\right)$ for all $i = 1, ..., N$.

2. For each pair $(X_k, Y_k) \in \mathcal{C}$, set $\lambda_k \triangleq \max \left\{ \hat{\lambda} : d\left(\hat{f}_{\mathcal{M}_{\hat{\lambda}}}(X_k), Y_{k\,\mathcal{M}_{\hat{\lambda}}}\right) \leq \alpha \right\}$.

3. Set $\lambda$ to be the $1 - \delta$ quantile of $\{\lambda_k\}_{k=1}^K$.

4. Define the final mask model as $\mathcal{M}_{\lambda}(X)_{[i]} \triangleq C\left(\widetilde{\mathcal{M}}(X)_{[i]}; \lambda\right)$.

**Output:** Calibrated uncertainty mask model $\mathcal{M}_{\lambda}$.

---

## 5 EXPERIMENTS

### 5.1 DATASETS AND TASKS

**Datasets** Two data-sets are used in our experiments:

Places365 (Zhou et al., 2017): A large collection of 256x256 images from 365 scene categories. We use 1,803,460 images for training and 36,500 images for validation/test.

Rat Astrocyte Cells (Ljosa et al., 2012): A dataset of 1,200 uncompressed images of scanned rat cells of resolution $990 \times 708$. We crop the images into $256 \times 256$ tiles, and randomly split them into train and validation/test sets of sizes 373,744 and 11,621 respectively. The tiles are partially overlapped as we use stride of 32 pixels when cropping the images.

**Tasks** We consider the following image-to-image tasks (illustrated in Figure 4):

Image Completion: Using gray-scale version of Places365, we remove middle vertical and horizontal stripes of 32 pixel width, and aim to reconstruct the missing part.

Super Resolution: We experiment with this task on the two data-sets. The images are scaled down to $64 \times 64$ images where the goal is to reconstruct the original images.

Colorization: We convert the Places365 images to grayscale and aim to recover their colors.

### 5.2 EXPERIMENTAL SETTINGS

**Image-to-Image Models** We start with training models for the above three tasks. Note that these models are not intended to be state-of-the-art, but rather used to demonstrate the uncertainty estimation technique proposed in this work. We use the same model architecture for all tasks: an 8 layer U-Net. For each task we train two versions of the network: (i) A simple regressor; and (ii) A conditional GAN, where the generator plays the role of the reconstruction model. For the GAN, the discriminator is implemented as a 4 layer CNN. We use the L1 loss as the objective for the regressor, and add an adversarial loss for the conditional GAN, as in Isola et al. (2017). All models are trained for 10 epochs using Adam optimizer with a learning rate of 1e-5 and a batch size of 50.

**Mask Model** For our mask model we use an 8 layer U-Net architecture for simplicity and compatibility with previous works. The input to the mask model are the measurement image and the predicated image, concatenated on the channel axis. The output is a mask having the same shape

as the predicted image with values within the range $[0, 1]$. The mask model is trained using the loss function (11) with $\mu = 2$, a learning rate of 1e-5 and a batch size of 25.

**Experiments**  We consider the L1, L2, SSIM and LPIPS as our divergence measures. We set aside $1,000$ samples from each validation set for calibration and use the remaining samples for evaluation. We demonstrate the flexibility of our approach by conducting experiments on a variety of 12 settings: (i) Image Completion: {Regressor, GAN} $\times$ {L1, LPIPS}; (ii) Super Resolution: {Regressor, GAN} $\times$ {L1, SSIM}; and (iii) Colorization: {Regressor, GAN} $\times$ {L1, L2}.

**Risk and Error Levels**  Recall that given a predicted image, our goal is to find a mask that, when applied to both the prediction and the (unknown) reference image, reduces the distortion between them to a predefined risk level $\alpha$ with high probability $\delta$. Here we fix $\delta = 0.9$ and set $\alpha$ to be the $0.1$-quantile of each measure computed on a random sample from the validation set, i.e. roughly $10\%$ of the predictions are already considered sufficiently good and do not require masking at all.

## 5.3 Competing Techniques for Comparison

**Quantile – Interval-Based Technique**  We compare our method to the quantile regression option presented in (Angelopoulos et al., 2022b), denoted by Quantile. While their calibrated uncertainty intervals are markedly different from the expected distortion we consider, we can use these intervals and transform them into a mask using (6). For completeness, we also report the performance of the quantile regression even when it is less suitable, i.e. when the underlying model is a GAN and when the divergence function is different from L1. We note again that for the sake of a fair comparison, our implementation of the mask model uses exactly the same architecture as the quantile regressor.

**Opt – Oracle**  We also compare our method with an oracle, denoted Opt, which given a ground-truth image computes an optimal mask by minimizing (10). We perform gradient descent using Adam optimizer with a learning rate of $0.01$, iterating until the divergence term decreases below the risk level $\alpha$. This approach is performed to each test image individually, thus no calibration needed.

**Comparison Metrics**  Given a mask $\mathcal{M}(X)$ we assess its performance using the following metrics: (i) Average mask size $s\big(\mathcal{M}(X)\big) \triangleq \frac{1}{N}\|\mathcal{M}(X) - \mathbb{1}\|_1$; (ii) Pearson correlation $Corr(\mathcal{M}, d)$ between the mask size and the full (unmasked) divergence value; and (iii) Pearson correlation $Corr(\mathcal{M}, \mathcal{M}_{opt})$ between the mask size and the optimal mask $\mathcal{M}_{opt}$ obtained by Opt.

## 5.4 Results and Discussion

We now show a series of results that demonstrate our proposed uncertainty masking approach, and its comparison with Opt and Quantile[2]. We begin with a representative visual illustration of our proposed mask for several test cases in Figure 2. As can be seen, the produced masks indeed identify sub-regions of high uncertainty. In the image completion task the bottom left corner is richer in details and thus there is high uncertainty regarding this part in the reconstructed image. In the colorization task, the mask highlights the colored area of the bus which is the most unreliable region since can be colorized with a large variety of colors. In the super resolution task the mask marks regions of edges and text while trustworthy parts such as smooth surfaces remain unmasked.

We present quantitative results in Table 1, showing that our method exhibits smaller mask sizes, aligned well with the masks obtained by Opt. In contrast, Quantile overestimates and produces larger masks as expected. In terms of the correlation $Corr(\mathcal{M}, d)$, our method shows high agreement, while Quantile lags behind. This correlation indicates a much desired adaptivity of the estimated mask to the complexity of image content and thus to the corresponding uncertainty. We provide a complement illustration of the results in Figure 3 in the Appendix. As seen from the top row, all three methods meet the probabilistic guarantees regarding the divergence/loss with fewer than $10\%$ exceptions, as required. Naturally, Opt does not have outliers since each mask is optimally calibrated by its computation. The spread of loss values tends to be higher with Quantile, indicating weaker performance. The middle and bottom rows are consistent with results in Table 1, showing that our approach tends to produce masks that are close in size to those of Opt; while Quantile

---

[2]Due to space limitations, we show more extensive experimental results in the Appendix, while presenting a selected portion of them here.

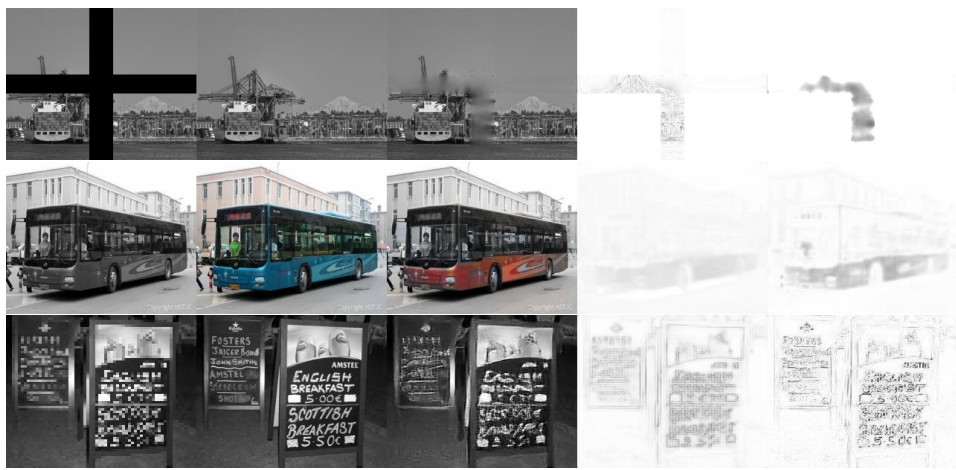

Figure 2: **Examples of conformal prediction masks.** The images from left to right are the measurement, ground-truth, model prediction, our calibrated mask trained with L1 loss and the ground-truth L1 error. Tasks are image completion (top), colorization (middle) and super resolution (bottom).

Table 1: Quantitative results. Arrows points to the better direction where best results are in **blue**.

| Network | Distance | $s(\mathcal{M})$ $(\downarrow)$ | | | $Corr(\mathcal{M}, d)$ $(\uparrow)$ | | $Corr(\mathcal{M}, \mathcal{M}_{opt})$ $(\uparrow)$ | |
|---|---|---|---|---|---|---|---|---|
| | | Opt | Ours | Quantile | Ours | Quantile | Ours | Quantile |
| Image Completion - Places365 | | | | | | | | |
| Regression | L1 | 0.09 | **0.10** | 0.15 | **0.89** | 0.78 | **0.89** | 0.76 |
| Regression | LPIPS | 0.01 | **0.01** | 0.20 | **0.54** | 0.51 | **0.89** | 0.77 |
| GAN | L1 | 0.09 | **0.09** | 0.14 | **0.95** | 0.85 | **0.94** | 0.80 |
| GAN | LPIPS | 0.01 | **0.01** | 0.08 | **0.31** | 0.24 | **0.50** | 0.23 |
| Super Resolution - Rat Astrocyte Cells | | | | | | | | |
| Regression | L1 | 0.24 | **0.26** | 0.28 | **0.99** | 0.54 | **0.95** | 0.88 |
| Regression | SSIM | 0.03 | **0.03** | 0.13 | **0.66** | 0.64 | **0.82** | 0.57 |
| GAN | L1 | 0.26 | **0.30** | 0.40 | **0.94** | 0.63 | **0.80** | 0.72 |
| GAN | SSIM | 0.03 | **0.03** | 0.13 | **0.79** | 0.63 | **0.83** | 0.63 |
| Super Resolution - Places365 | | | | | | | | |
| Regression | L1 | 0.30 | **0.36** | 0.39 | **0.99** | 0.97 | **0.95** | 0.94 |
| Regression | SSIM | 0.10 | **0.23** | 0.48 | **0.89** | 0.85 | **0.94** | 0.84 |
| GAN | L1 | 0.37 | **0.38** | 0.47 | **0.97** | 0.81 | **0.95** | 0.67 |
| GAN | SSIM | 0.10 | **0.12** | 0.51 | **0.86** | 0.81 | **0.92** | 0.86 |
| Colorization - Places365 | | | | | | | | |
| Regression | L1 | 0.27 | **0.37** | 0.40 | **0.68** | 0.43 | **0.57** | 0.46 |
| Regression | L2 | 0.18 | **0.37** | 0.38 | **0.57** | 0.30 | **0.60** | 0.48 |
| GAN | L1 | 0.27 | **0.38** | 0.40 | **0.58** | 0.40 | **0.60** | 0.52 |
| GAN | L2 | 0.18 | **0.36** | 0.38 | **0.42** | 0.28 | **0.59** | 0.49 |

produces larger, and thus inferior, masked areas. We note that the colorization task seem to be more challenging, resulting in a marginal performance increase for our method compared to Quantile.

## 6 CONCLUSIONS

Uncertainty assessment in image-to-image regression problems is a challenging task, due to the implied complexity, the high dimensions involved, and the need to offer an effective and meaningful visualization of the estimated results. This work proposes a novel approach towards these challenges by constructing a conformal mask that visually-differentiate between trustworthy and uncertain regions in an estimated image. This mask provides a measure of uncertainty accompanied by an statistical guarantee, stating that with high probability, the divergence between the original and the recovered images over the non-masked regions is below a desired risk level. The presented paradigm is flexible, being agnostic to the choice of divergence measure, and the regression method employed.

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

## A  APPENDIX: QUANTITATIVE RESULTS

Here we provide additional graphical results to complement the quantitative results of Table 1.

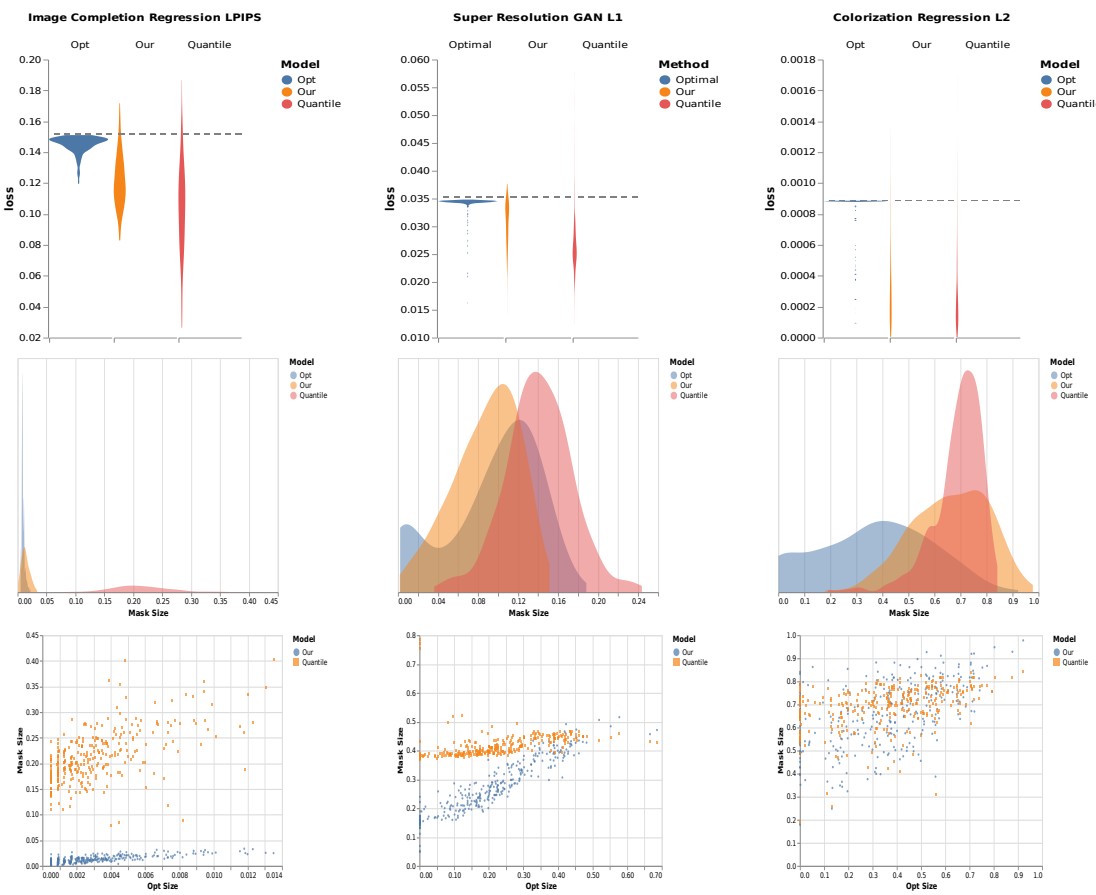

Figure 3: **Quantitative results.** (Top) Distribution of divergence values after masking, (middle) histograms of mask sizes, and (bottom) correlation with mask sizes obtained by Opt.

## B    APPENDIX: TASK ILLUSTRATIONS

In the Figure below we illustrate the three tasks we experiment with in this work.

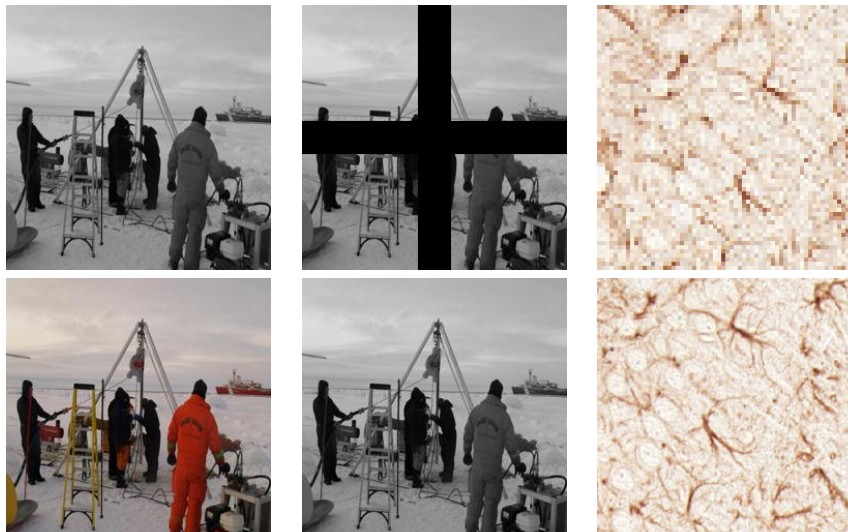

Figure 4:   The three tasks we experimented with:  1) Image Colorization on the left column 2) Gray-scale Image Completion on the middle column, and 3) Super Resolution on the right column.

## C    APPENDIX: MORE RESULTS

Here we provide the results of all 12 settings explored in our work, corresponding to three inverse problems, two regression techniques and two metrics per each, along the following breakdown:

- Image Completion: {Regressor, GAN} $\times$ {L1, LPIPS};
- Super Resolution: {Regressor, GAN} $\times$ {L1, SSIM}; and
- Colorization: {Regressor, GAN} $\times$ {L1, L2}.

We start in Figures 5-7 with the obtained distributions of masked distortion values obtain by our method, Opt, and Quantile.  The goal here is to show that all three methods meet the required divergence condition with exceptions which do not surpass the destination probability $\delta = 0.1$.

Figures 8-10 present histograms of the obtained mask sizes for the three methods. The goal is to get minimal areas in these masks, so as to keep most of the image content unmasked.

Figures 11-13 conclude these results by bringing graphs showing the inter-relation between the Opt mask size and the ones given by our technique and Quantile.

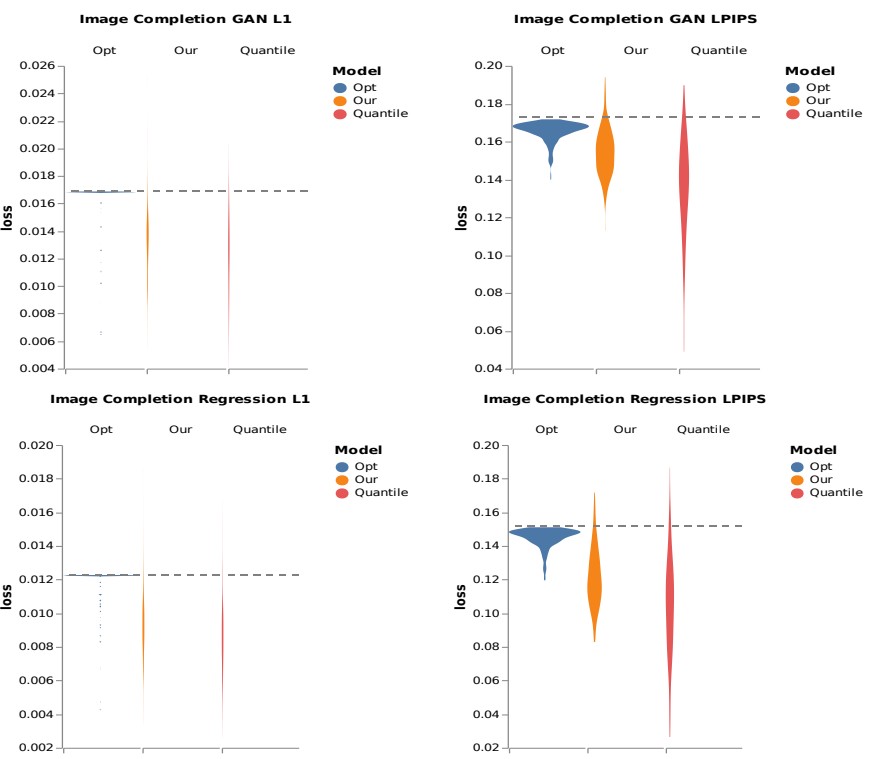

Figure 5: **Image Completion.** Distribution of the masked divergence values versus the chosen risk level (shown as a horizontal dashed line) for the three tested methods - Opt, Quantile and ours.

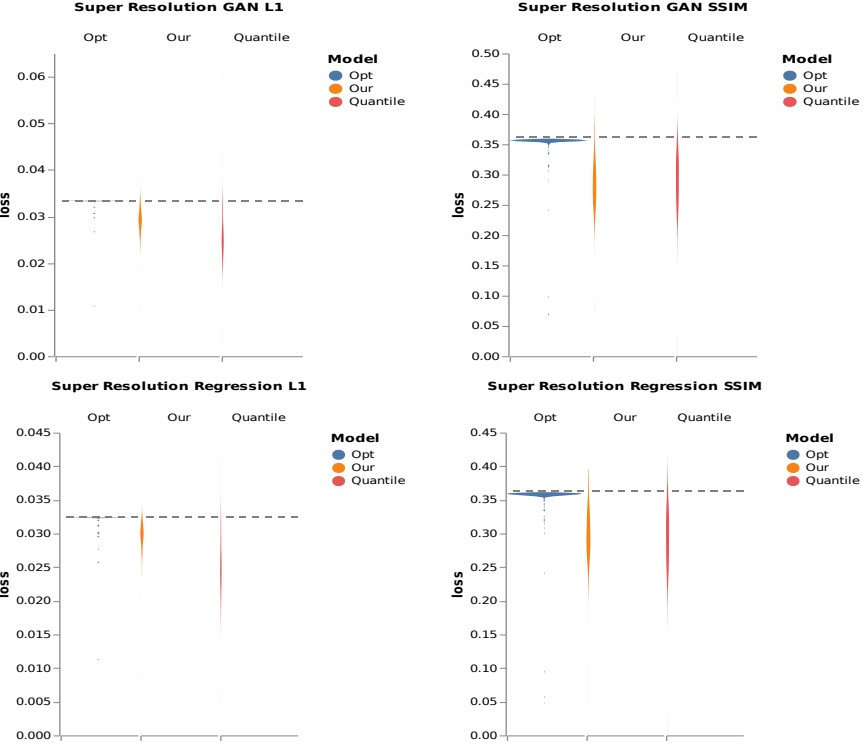

Figure 6: **Super Resolution.** Distribution of the masked divergence values versus the chosen risk level (shown as a horizontal dashed line) for the three tested methods - Opt, Quantile and ours.

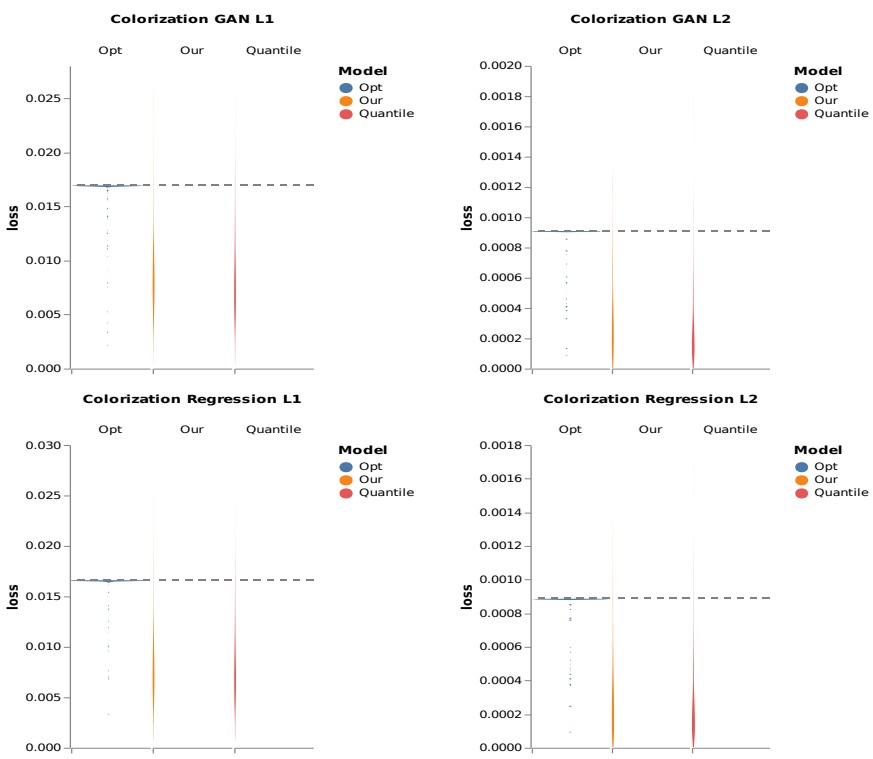

Figure 7: **Colorization.** Distribution of the masked divergence values versus the chosen risk level (shown as a horizontal dashed line) for the three tested methods - Opt, Quantile and ours.

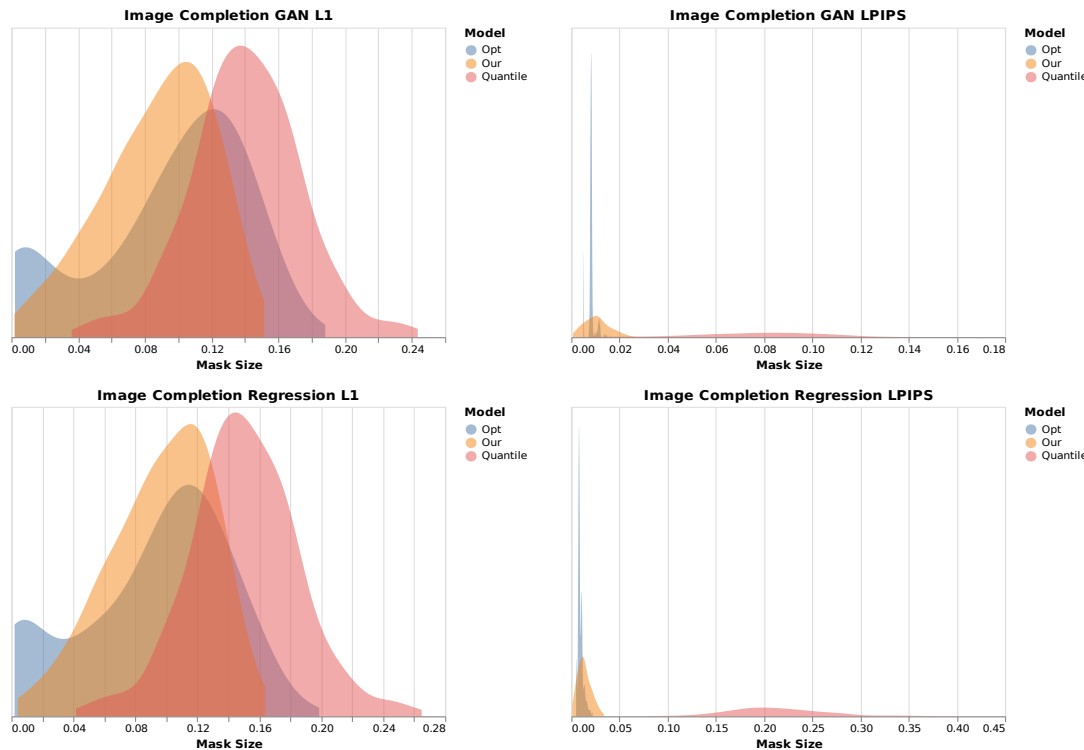

Figure 8: **Image Completion.** Histograms of the calibrated mask sizes for the three tested methods - Opt, Quantile and ours.

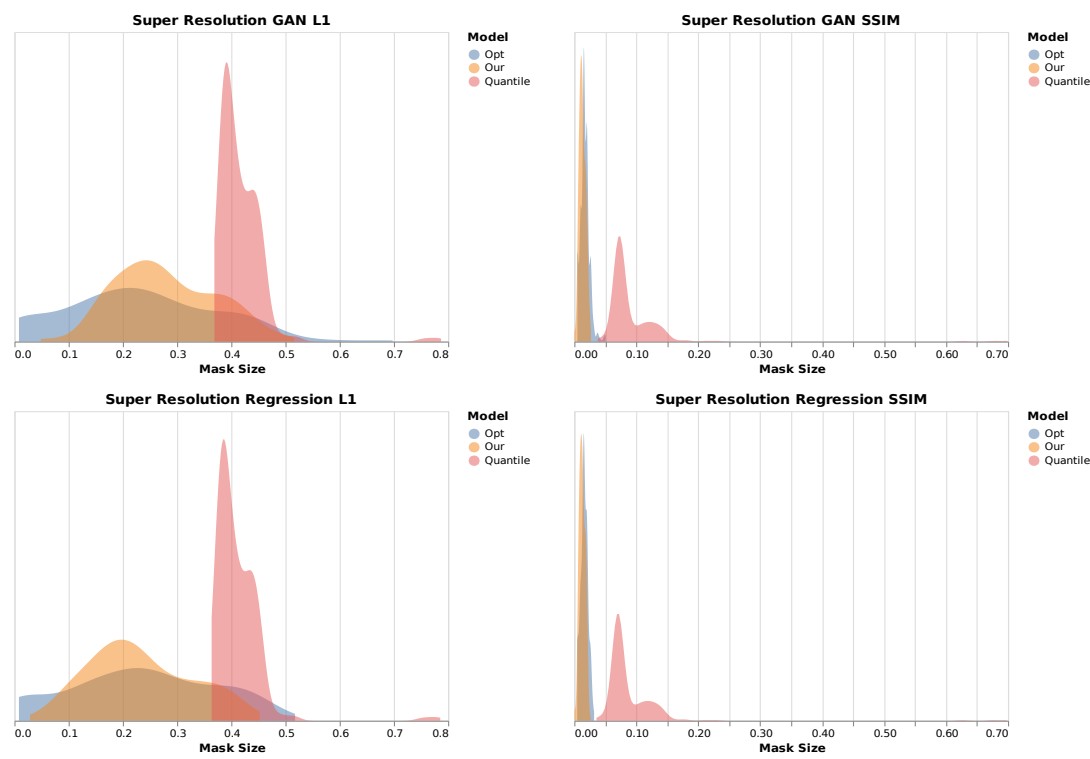

Figure 9: **Super Resolution.** Histograms of the calibrated mask sizes for the three tested methods - Opt, Quantile and ours.

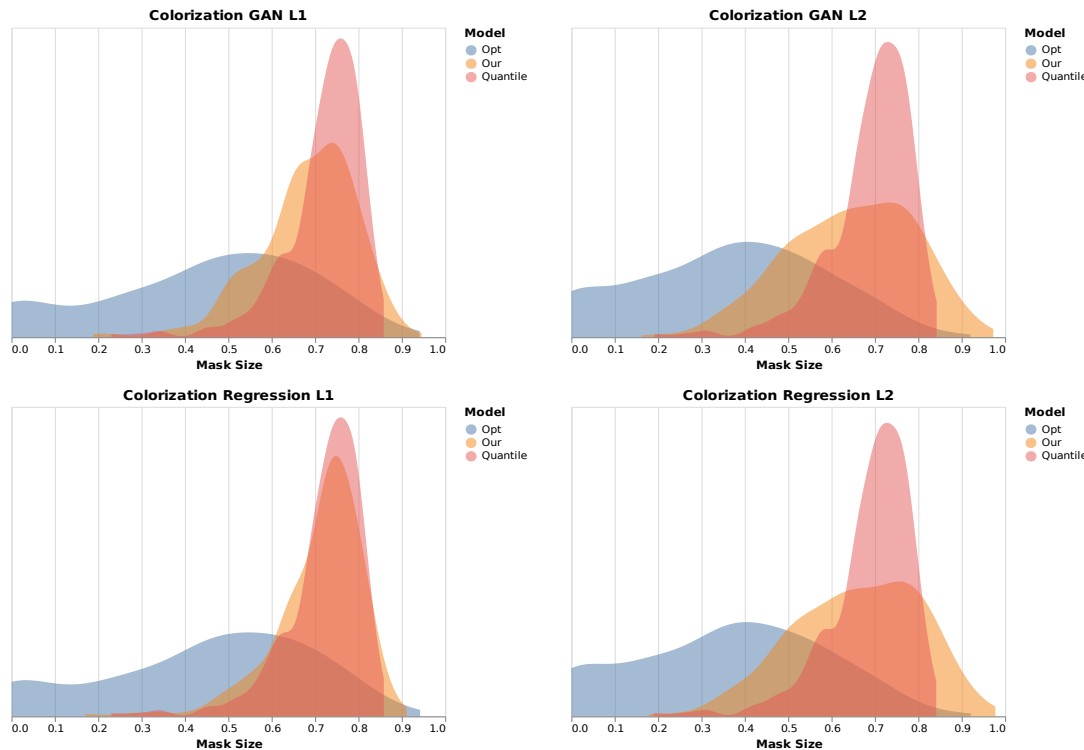

Figure 10: **Colorization.** Histograms of the calibrated mask sizes for the three tested methods - Opt, Quantile and ours.

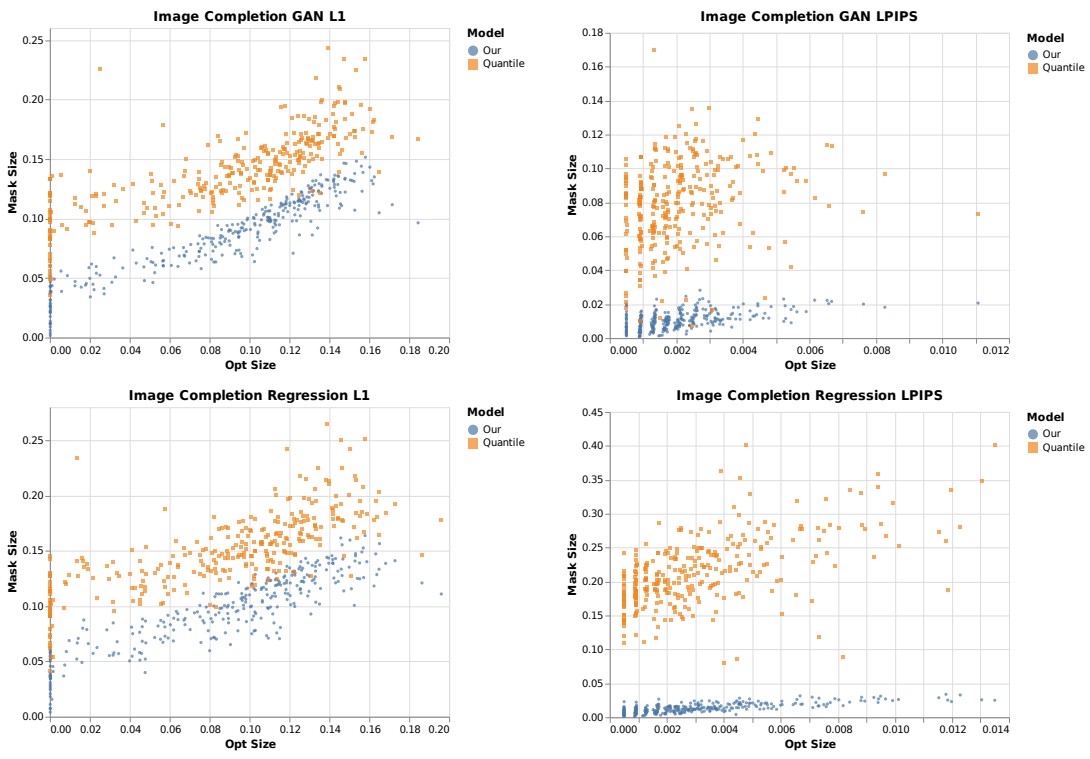

Figure 11: **Image Completion.** Correlation between the mask sizes produced by our method and Qunatile versus the mask size obtained by Opt.

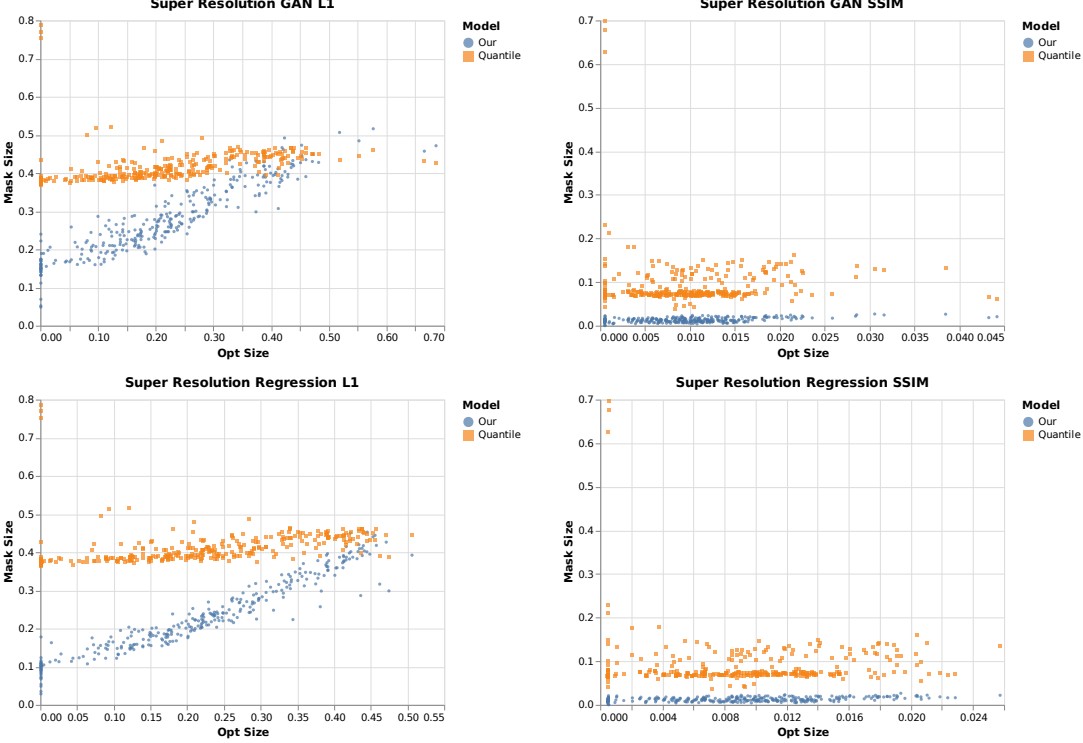

Figure 12: **Super Resolution.** Correlation between the mask sizes produced by our method and Qunatile versus the mask size obtained by Opt.

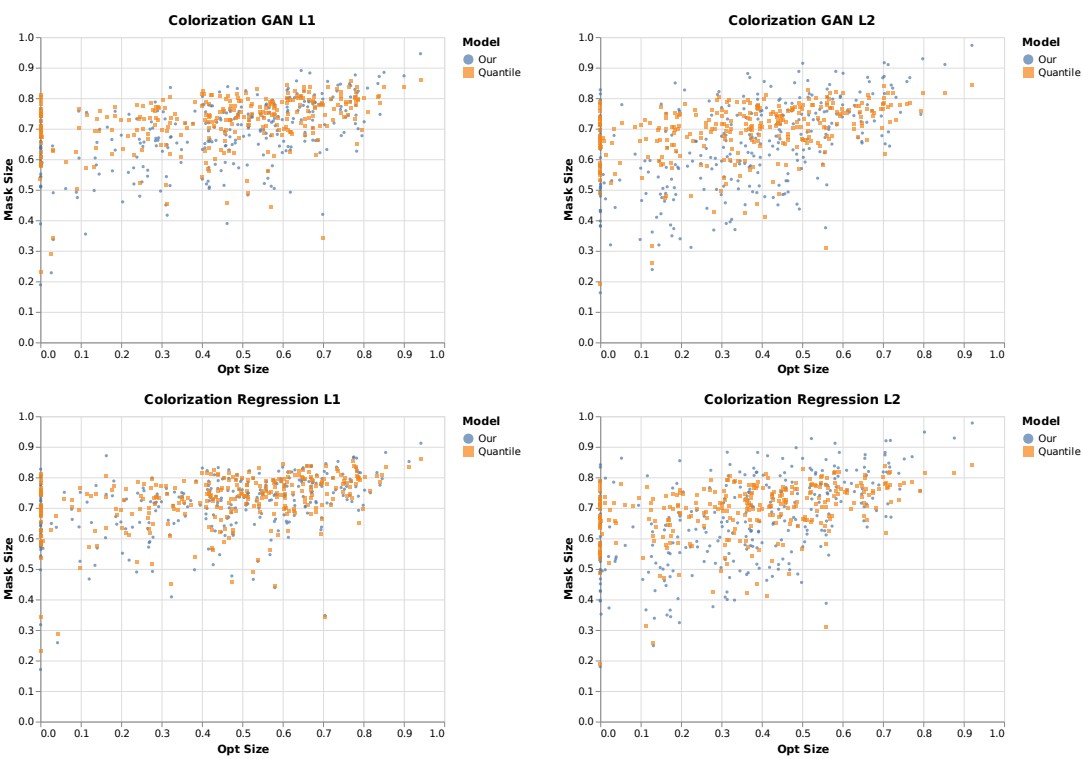

Figure 13: **Colorization.** Correlation between the mask sizes produced by our method and Qunatile versus the mask size obtained by Opt.

