# OpenReview forum: "Conformal Prediction Masks: Visualizing Uncertainty in Medical Imaging"
_ICLR.cc/2023/Workshop/TML4H — ICLR 2023 Workshop TML4H Poster_

### Official Review · Reviewer_nn6z · 2023-03-01
**Interesting methods but Insufficient Experimental Validation**

**Rating:** 5
**Confidence:** 3

**Review:**

This paper proposes a novel approach for uncertainty quantification and visualization in image regression tasks, using conformal prediction masks to identify reliable regions and areas of high uncertainty. The theoretical framework is well-presented, including mathematical definitions and statistical guarantees. The proposed approach is evaluated on various image recovery tasks, which demonstrates its effectiveness.

Pros:

1. Good theoretical clarification, including mathematical definitions and proofs of the statistical guarantees.
2. The construction of an uncertainty mask for regression task from Conformal prediction is interesting.
3. The approach is evaluated on different regression tasks, demonstrating its versatility.

Cons:

1. The evaluation should be extended to more medical datasets, as the current evaluation on rat cells may not be sufficient to fully validate the approach's performance.
2. To indicate the superiority of the proposed uncertainty method, the authors should compare it with other Bayesian-based uncertainty quantification techniques.
3. The evaluation metrics could be further considered, as it is unclear why the average mask size is included as an evaluation metric. The authors should double-check the tables and figures, such as in Table 1, where it is confusing why "ours" performs worse in s(M) but is marked in blue.

---

### Meta-Review · Area_Chair_9p8Q · 2023-03-06

**Recommendation:** Accept (Poster)
**Confidence:** 3

**Metareview:**

The paper presents a new masking-based uncertainty quantification and visualization approach, which is evaluated on image colorization, image completion and super-resolution tasks, achieving good performance.

Pros:
1)	The proposed method is novel and can contribute to this research field. It was clearly presented with clear mathematical definitions and proofs.
2)	The evaluation of the proposed method is comprehensive, including different computer vision tasks, demonstrating its efficacy.
3)	The paper is well organized and easy to follow.
Cons:
1)	Please give the definition of the evaluation metric Corr(,).
2)	I suggest the authors add “Eq.” Before the citation of equations in the manuscript (i.e., Eq. (1)).
3)	The limitations and future directions should be briefly discussed in the manuscript.
4)	Comparison with some bayesian uncertainty quantification methods is needed.